# The Development and Progress of the UWB Physical Layer

**DOI:** 10.3390/mi14010008

**Published:** 2022-12-21

**Authors:** Ziteng Lv, Xin Zhang, Dongdong Chen, Di Li, Xianglong Wang, Tianlong Zhao, Yintang Yang, Yanbo Zhao, Xin Zhang

**Affiliations:** 1School of Microelectronics, Xidian University, Xi’an 710071, China; 2Guangzhou Institute of Technology, Xidian University, Guangzhou 510555, China

**Keywords:** UWB, physical layer, encoder, pulse generator, receiver, decoder

## Abstract

Ultra-wideband (UWB) technology has been applied in many fields, such as radar and indoor positioning, because of its advantages of having a high transmission rate, anti-multipath interference, and good concealment. In the UWB physical layer, the transmitting link, including an encoder and a pulse generator, is used to improve the anti-interference ability of the signal, while the receiving link, including a receiver and a decoder, can correct the error signal. Therefore, the performance of the UWB physical layer can obviously affect the speed and quality of UWB signal transmission. In this paper, the structure and performance of the codec and transceiver of the UWB physical layer are introduced and compared. In addition, some typical architectures and features are summarized and discussed, which provides a valuable reference and suggestions for the design of the UWB physical layer. Finally, the outlook of the UWB physical layer is presented: its development direction mainly includes high speed, low power consumption, and fewer hardware resources.

## 1. Introduction

Ultra-wideband (UWB) technology has attracted attention for its advantages such as its high transmission rate, anti-multipath interference, and good robustness [1,2,3]. Due to the advantages of requiring low power, having a small size, and strong penetration, the UWB system has been widely applied in many fields such as indoor positioning [4,5,6,7,8], wireless body area network [9,10,11,12], and radar [13,14,15]. It has become one of the most important technologies in wireless communication research. Specifically, the UWB physical layer can achieve data transmitting, which is an important part of the UWB system. Therefore, the investigation of the UWB physical layer is essential.

The UWB physical layer transceiver link structure is shown in Figure 1 [16]. Reed Solomon (RS) encoders are used for payload bits from the physical layer to increase its antijamming ability [17]. Then, a convolutional encoder is used to sort the RS encoded data. A scrambler is inserted into the transmitting link to increase the antijamming capability [18]. The encoded data are emitted by a pulse generator [19,20]. After receiving the information, the receiver demodulates the data, and then decodes the data through a Viterbi decoder and an RS decoder, respectively, to obtain the information [21,22,23,24].

The performance of the UWB physical layer determines the speed and accuracy of data transmission, so its investigation is essential to improve the transmitting performance [25]. However, the traditional UWB physical layer cannot satisfy the high requirements of speed and accuracy. Firstly, the coding speed of traditional encoders is slow when large amounts of data are encoded. Secondly, a pulse generator can interfere with the narrow band systems. Additionally, the receiver can be interfered with in a multipath environment. In addition, the complex algorithm can increase hardware resources and power consumption when decoding data. Therefore, the UWB physical layer transceiver link faces the following challenges:(1)High-speed encoders are required to deal with large amounts of encoded data;(2)Interference between UWB and other narrowband systems should be reduced;(3)The anti-multipath capability of the receiver should be improved;(4)A low-complexity algorithm is needed for the decoder to reduce hardware resources.

In this paper, the development and prospects of the UWB physical layer transceiver are reviewed to provide valuable references for designing a high-performance UWB physical layer. Section 2 introduces the encoder module of the UWB transmitting link. The UWB pulse generator module is introduced in Section 3. Section 4 introduces the receiver module. In Section 5, the decoder module of the UWB receiver link is presented. Finally, a summary and outlook of the UWB physical layer are presented.

## 2. Encoder

The encoder is used to add a check bit for data, which can improve its anti-interference ability in channel transmission. In a UWB transmitting link, the encoder module is composed of an RS encoder, a convolution encoder, and a scrambler.

### 2.1. RS Encoder

An RS encoder is widely used in wireless communication, digital broadcasting, television, and deep space exploration. The speed of a traditional RS encoder is slow, and it can be blocked when encoding a large amount of data, which can reduce the encoding efficiency. In order to deal with this problem, Ren et al. [26] proposed a bit-parallel multiplication RS encoder based on a dual basis, as shown in Figure 2a. All multiplication and addition operations are applied to the dual-basis encoder, and a pipeline structure is adopted in the proposed algorithm, which can greatly increase the encoder efficiency. In addition, Mohamed et al. [27] proposed a linear feedback shift register (LFSR) RS encoder based on a vector string/parallel divider, as shown in Figure 2b. Multiple division operations can be performed in parallel in the proposed architecture. Compared with a traditional RS encoder with a polynomial length of 17, the proposed structure can improve throughput by about 83.3%.

Generally, traditional RS encoders occupy a large area and heavy hardware resources. Jitawutipoka et al. [28] proposed a Galois domain multiplier based on subexpression sharing to reduce hardware resources, which can reduce the number of exclusive OR (XOR) gates. Moreover, Wu et al. [29] optimized the multiplication algorithm by logical algebra to further decrease the number of XOR gates in the adder. For an RS (255,239) encoder, the number of total XOR gates without the optimization algorithm is 366. However, those in [28,29] are 276 and 246, respectively.

In order to solve the problem of different encoder rates in different channels, Lee et al. [30] proposed an RS encoder with extra memory, as shown in Figure 2c. The proposed architecture can provide adaptive bit rates according to the characteristics of irregular channels, which can be implemented in an upstream modulation Gbps cable transmission system.
Figure 2(**a**) Architecture of dual basis RS encoder. (Reprinted from [26], Copyright 2009, with permission from IEEE). (**b**) Circuit implementation of the vector string/parallel divider RS encoder. (Reprinted from [27], Copyright 2019, with permission from Elsevier). (**c**) Architecture of variable-rate RS encoder. (Reprinted from [30], Copyright 2011, with permission from Springer).
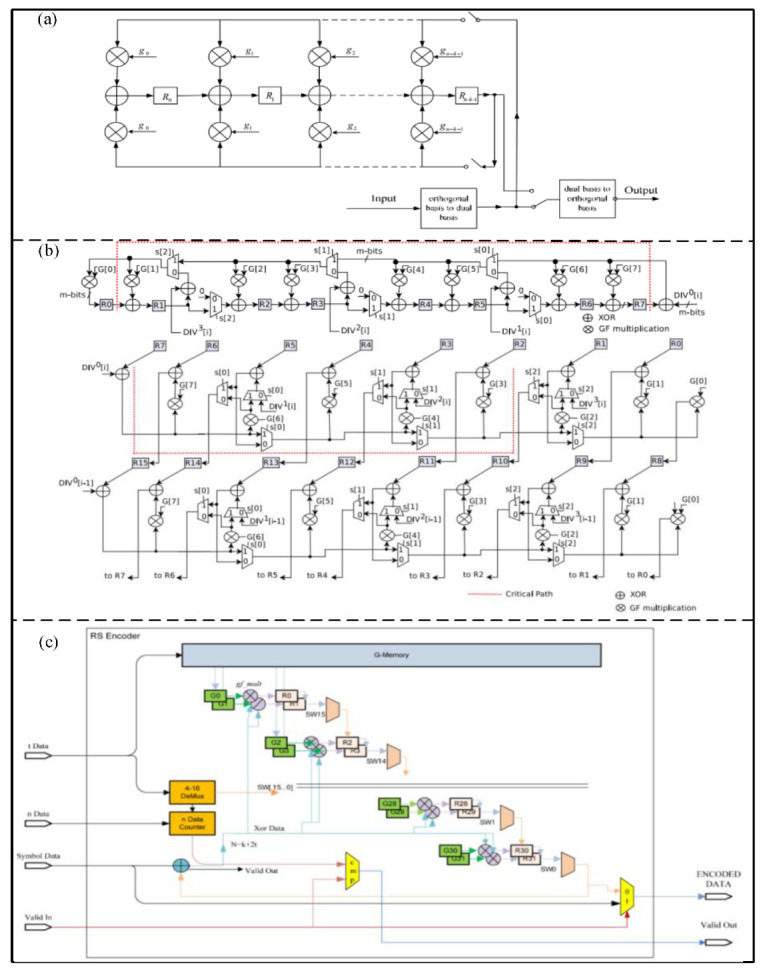



### 2.2. Convolutional Encoder

A convolutional encoder is used to sort the RS encoded data and suppress the spectrum [31]. In recent years, the pseudo-chaotic time hopping (PCTH) technique has been used in convolutional coders to suppress spectral lines and reduce the bit error rate (BER). Villarreal-Reyes et al. [32] proposed a new binary to m-base maximum free distance convolution code for time-hopping (TH) impulse radio (IR) UWB systems. Compared with the PCTH algorithm, the proposed algorithm has better hard and soft decision bit error rates, as shown in Figure 3a. However, the proposed algorithm cannot suppress the spectral lines in power spectral density (PSD), and the input of the encoder cannot be completely uniform binary numbers. In order to mitigate the above disadvantages, Villarreal-Reyes et al. [33] improved the algorithm again, and proposed a convolution encoder suitable for binary phase shift keying (BPSK) and quaternary bi-orthogonal pulse position modulated (Q-BOPPM) TH-IR UWB systems. In this encoder, the power spectral density cannot exceed the spectral limit; even the input of the encoder consists of nonuniformly distributed binary numbers, with the result shown in Figure 3b. In addition, Aldo et al. [34] proposed a spectral line free (SLF) convolution encoded IR-UWB over a fiber system with higher transmission power to increase the transmission distance and reduce the BER.

### 2.3. Scrambler

The anti-interference capability of transmitting data is an important research direction in wireless communication. Scramblers are used to whiten baseband data to reduce error and bit error rates during transmission. In order to reduce the PSD of UWB, Mo et al. [35] proposed a scrambler with a two-layer LFSR to increase the randomness of the scrambler. As shown in Figure 4, the proposed scrambler can increase its randomness in the Y direction. Compared with LFSR-15 with four seeds, the proposed structure can reduce by about 32 dB the UWB physical layer PSD. In addition, Kouassi et al. [36] proposed a UWB transmitter with a random pulse-width scrambling method. After analyzing and optimizing with a genetic algorithm, the result shows that the informationless scrambler can improve spectrum utilization and antijamming ability.

### 2.4. Comparison and Discussion

The encoder style, scope, and effect of three different encoders are presented in Table 1. These three encoders are used to increase the anti-interference to reduce the bit error rate. The coding manners of RS and convolutional encoders are both forward error correction codes. The application ranges of these three codes are different. An RS encoder is used to encode the PSDU, the convolution encoder is used to sort the PSDU, and scramblers act on preamble codes.

## 3. Pulse Generator

### 3.1. Low Spectral Interference Generator

A pulse generator is used to load the encoded data onto the carrier, which is essential in the UWB physical layer. The UWB system may conflict with other narrowband systems, so many circuits have been proposed to reduce PSD [37,38]. Dong et al. [39] proposed a 0.18 μm UWB pulse generator with a pulse oscillator structure and two optional channels for 3–5 GHz, which includes a control circuit, a pulse generation core, an inverter, and a balun, as shown in Figure 5a. The on-chip balun with high-pass characteristics shows high suppression towards PSD in the Global Positioning System (GPS) band. The measurement results showed that the PSD suppression was about 19 dB. Additionally, Sim et al. [37] proposed a low-power, high-peak-value UWB pulse generator based on the 0.18 μm process for 6–10 GHz, which consisted of a pulse generator, a pulse-shaping filter, and a pulse oscillator, as shown in Figure 5b. The pulse-shaping filter was used to make the spectrum comply with Federal Communications Commission (FCC) regulations. For the GPS band at 0.96–1.61 GHz, the proposed pulse-shaping filter can suppress the spectral component by more than 34 dB. As shown in Figure 5c, Zhao et al. [38] proposed a CMOS UWB pulse generator for 3–5 GHz with on–off keying (OOK) modulation, in which a new push-and-pull-integrating narrow triangular pulse generator was designed to reduce the common mode interference and static current. In addition, a new on–off voltage-controlled ring oscillator (VCRO) with a complementary switch mode was proposed, which can reduce power consumption by avoiding generating base-band energy. The results show that the proposed structure can suppress the sidelobes by more than 20 dB. Based on the 90 nm CMOS process, Hedayati et al. [40] proposed a fully integrated analog UWB pulse transmitter with a BPSK modulation, as shown in Figure 5d, which can co-exist with the IEEE 802.11a system. The measurement results show that the generator can produce a tunable notch with a 30 dB attenuation in the narrowband system. Based on the 0.18 um CMOS process, Gunturi et al. [41] proposed a 250 Mb/s data rate IR-UWB transmitter, composed of a pulse generator and a current-reused power amplifier, as shown in Figure 5e. The experimental result shows that the peak PSD power is −42 dBm.

### 3.2. Optional Channel Generator

Due to the different UWB frequency bands of different countries, many pulse generators are only suitable for a specific frequency band. In order to solve this problem, Choi et al. [42] proposed an all-digital pulse generator based on the 0.13 μm CMOS technology, consisting of a delay line structure, as shown in Figure 6a. The center frequency and the bandwidth of the proposed generator are digitally controlled to cover three channels at 3.1–4.8 GHz. Based on the 65 nm CMOS process, Na et al. [43] proposed an all-digital UWB pulse generator with three optional channels. An all-digital oscillator (ADO) containing a delay line and an edge synthesizer was used to generate a UWB baseband signal. Compared to the architecture proposed by Choi, the architecture proposed by Na can achieve lower power consumption. In addition, Oliver et al. [44] proposed a UWB pulse generator with 14 frequency bands covering the whole UWB bandwidth. As shown in Figure 6b, the proposed pulse generator includes a wideband T/R switch, radio frequency balun, and a full phase-locked loop (PLL) filter assembly, and was fabricated via the 0.13 μm SiGe BiCMOS process.

### 3.3. Low Power Consumption Generator

Low power consumption is also an important area of the UWB physical layer [45]. Based on the 0.18 um CMOS process, Zheng et al. [46] proposed a burst mode super-regenerative low-power UWB generator with an OOK modulation, which can be applied in a wireless body area network, as shown in Figure 7a. The proposed generator can restore the received signal and reduce the post-stage amplifier, and its power consumption is 671 pJ/pause. In order to achieve low power consumption, Mercier et al. [47] proposed a low-power all-digital UWB pulse generator with a BPSK + PPM modulation based on the 90 nm CMOS process, as shown in Figure 7b. The proposed generator uses a simple single-ended ring oscillator and a digital output buffer to synthesize the pulse, and the measured energy consumption is 17.5 pJ/pause. As shown in Figure 7c, Bourdel et al. [48] proposed a low-power pulse response filter OOK-modulated UWB pulse generator, fabricated by the 0.13 um CMOS process. The edge synthesizer is used to excite an integrated bandpass filter, and the proposed generator energy consumption is 9 pJ/pause. In addition, Shen et al. [49] proposed a UWB pulse generator based on the 0.18 μm CMOS process with low peak power consumption, as shown in Figure 7d. In order to decrease the peak current and maintain the waveform of the generated UWB pulse signal, a slow-charging and fast-discharging method was adopted in the proposed generator to increase the duration of the pulse peak current. The measurement results show that the minimum power consumption is 5 pJ/pause. Based on the 0.18 um CMOS process, Radic et al. [50] proposed a low-power IR-UWB transmitter, consisting of a controllable pulse generator, a switchable tunable oscillator, a driver, and a pulse-shaping filter, as shown in Figure 7e. The experimental results show that the minimum power consumption is only 3 pJ/pulse.

### 3.4. Comparison and Discussion

The summary of the UWB pulse generator is shown in Table 2. The process, bandwidth, power consumption, modulation mode, circuit area, etc., are systematically compared and discussed. Obviously, the 0.18 μm CMOS process is the most mainstream process at present. Meanwhile, some more advanced processes, such as 90 nm CMOS and 65 nm CMOS, have also been used in the fabrication of a pulse generator, which can reduce the power consumption and die photograph area. As shown in Table 2, the pulse oscillator, delay lines, and analog circuits are major architectures in a pulse generator. BPSK modulation can suppress spectral line problems and has a low bit error rate. Additionally, OOK modulation shows low bit error rate characteristics with a simple structure. Therefore, most pulse generators adopt the BPSK and OOK modulation. The pulse generators in [43,44,45] provide optional channels, but their −10 dB bandwidth is only 0.5 GHz, which is less than other architectures. Compared with other pulse generators, the tunable pulse in [51] leads to significantly low power consumption, 3 pJ/pause. The generator proposed by Shen et al. [50] can operate at a high pulse repetition rate (PRR) of 1 Gpps, which is the highest.

## 4. Receiver

The pulse signal is transmitted through an environmental channel, which is demodulated and filtered for noise by the receiver. Rake and analog front-end receivers applied in the UWB physical layer are introduced.

### 4.1. Rake Receiver

A rake receiver is a key technology of a spread-spectrum communication system, and was proposed by Price and Green in 1958 [51]. The multipath signal energy is processed by the rake receiver to improve the signal-to-noise ratio (SNR) of the signal and reduce the fading probability [52]. Rake receivers are divided into multipath collection and combination. In order to implement a multipath collection strategy, the ideal rake receiver is used to analyze all multipath components which can be called all-rake (A-rake). The A-rake has the best BER performance; however, its architecture is not widely used because of the high complexity.

In order to balance complexity and performance, two UWB rake receivers with low complexity were proposed by Cassioli et al. [53], a partial rake (P-rake) and selective rake (S-rake). A P-rake has the lowest complexity because it only combines the multipath components that arrive to the receiver first, while the instantaneous strongest multipath component is received by the S-rake. As shown in Figure 8a, the performance of the S-rake and P-rake is similar at a high-frequency channel, while the S-rake significantly outperforms the P-rake at a low-frequency channel. In addition, adaptive P-rake and S-rake receivers were proposed by Doukeli [54], and can be applied in the UWB physical layer. In the proposed receivers, the number of combined paths can be reduced by comparing the SNR quality of each path. As shown in Figure 8b, the performance of these proposed rake structures is higher than that of S-rake and P-rake.

Different combination methods are used by the rake receivers in the multipath combination strategy. Maximum ratio combination (MRC) is known as an optimal linear combination technology, which combines all paths with different weights. The combination weights can be selected by a minimum mean square error (MMSE) equalizer for better performance. A receiver hardware architecture that combines a rake and MMSE equalizer was proposed by Eslami [55]. Rake and equalizer structures are used to combat intersymbol interference. The performance of the proposed receiver relationship with the number of equalizer taps and rake fingers is studied by a semi-analytical approach and Monte Carlo simulation. As shown in Figure 8c, the number of rake fingers is the main factor that improves the performance of the system at low SNR, while the number of equalizer taps is more important at high SNR. In addition, a new adaptive rake MMSE receiver architecture based on the recursive least squares algorithm was proposed by Kang et al. [56]. The MMSE method is used by the proposed architecture to combine multipath components for eliminating narrowband interference. As shown in Figure 8d, the MMSE can achieve better performance than the traditional MRC and equal-gain combining (EGC) receivers. The performance of the proposed receiver is the best in CM1 (line-of-sight channel), and the worst in CM4 (non-line-of-sight channel).
Figure 8(**a**) Functional between BEP and normalized SNR at rake output of A−rake, S−rake, and P−rake in LF channel. The BEP vs. Eb/N0 for P−Rake and S−Rake in the HF channel model with the full transmission bandwidth. (Reprinted from [53], Copyright 2007, with permission from IEEE). (**b**) BEP for the case of the LF channel model. (Reprinted from [54], Copyright 2012, with permission from Springer). (**c**) UWB rake−MMSE−equalizer structure and performance of UWB rake−MMSE−receiver for different number of equalizer taps and rake fingers. (Reprinted from [55], Copyright 2005, with permission from IEEE). (**d**) BER comparison of three combining methods for Rake in CM3 and the BER performance of the proposed Rake−MMSE receiver under CM1−CM4 channel models. (Reprinted from [56], Copyright 2016, IEEE).
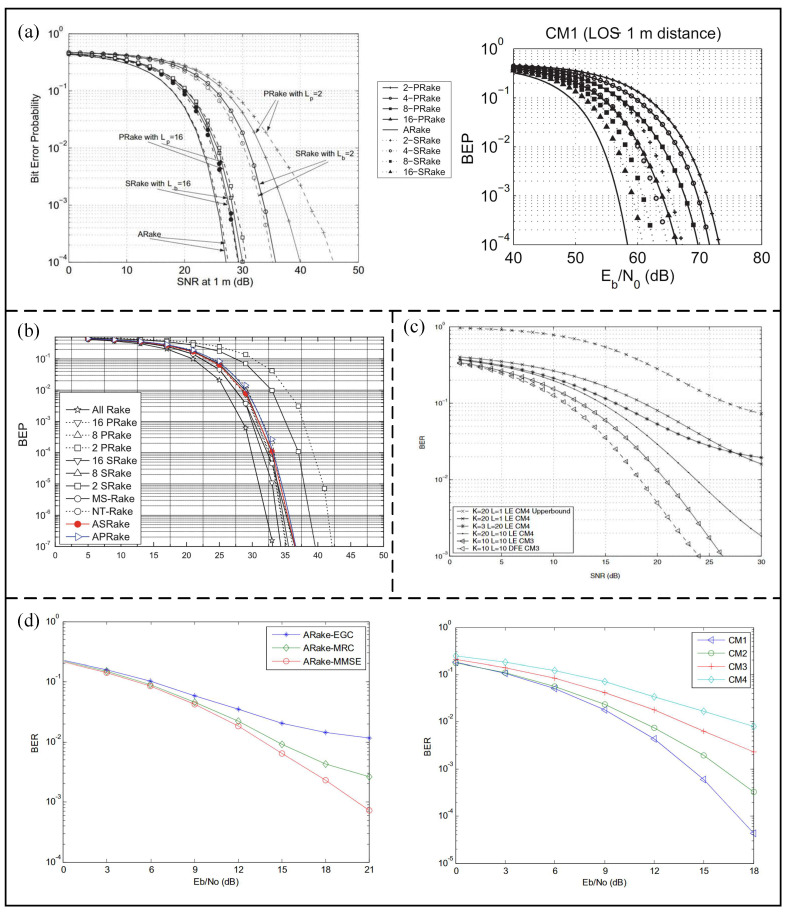



### 4.2. Analog Front-End Receiver

The received signal is amplified and digitized by the analog front-end receiver. Generally, the receiver consists of a low-noise amplifier, an analog-to-digital converter, and a digital signal processing unit.

In order to achieve low power consumption and high-data-rate communication, Medi et al. [57] proposed an ultra-wide band receiver implemented in 0.18 μm CMOS technology operating at a 3.25–4.75 GHz band and pulse-based, as shown in Figure 9a. Using the proposed receiver, the UWB signal can be effectively digitized and its robustness can be increased. When running at a data rate of 1 Gbps, the proposed receiver consumes 98 pJ/b in the receive mode. In addition, Ryckaert et al. [58] proposed a low-power pulse radio UWB receiver that can be applied at a low data rate. As shown in Figure 9b, the quadrature analog correlation architecture is used in the proposed receiver, which has low energy consumption because of reducing the ADC sampling speed. At the 20 Mpulses/s pulse rate, a 16 mA power consumption can be achieved by the proposed receiver implemented in the 0.18 μm CMOS.

In a narrowband interference channel, Anis et al. [59] proposed a UWB receiver architecture, which can extract effective signals by narrow-band-pass filters. The narrow-band-pass filter consists of a super-regenerative receiver (SRR), as shown in Figure 9c. The receiver has been implemented in 0.18 μm CMOS with a power consumption of 2.6 mW.

In order to reduce the power consumption and layout area, Terada et al. [60] proposed a CMOS UWB-IR receiver architecture with 0.18 μm CMOS technology, as shown in Figure 9d. The intermittent operation in the proposed architecture can reduce the power consumption mainly caused by the clock correlator and differential low-noise amplifier. The power consumption of the proposed receiver is only 1 mW when the differential low-noise amplifier works intermittently through the bias switch. Based on the 0.13 μm CMOS process, Helleputte et al. [61] proposed an integrated ultra-low-power analog front-end architecture for UWB pulse radio receivers, as shown in Figure 9e. A local oscillator is adopted to reduce the power consumption of pulse correlation, and the proposed receiver has a power consumption of 2.7 mW at a 39.0625 Mpulses/s pulse rate.
Figure 9(**a**) Frequency channelized receive and chip micrograph. (Reprinted from [57], Copyright 2008, with permission from IEEE). (**b**) Quadrature analog correlating architecture and receiver chip mi-crophotograph. (Reprinted from [58], Copyright 2007, with permission from IEEE). (**c**) Block dia-gram of Super regenerative receiver and die photograph for UWB receiver. (Reprinted from [59], Copyright 2007, with permission from IEEE). (**d**) Chip microphotograph. (Reprinted from [60], Copyright 2006, with permission from IEEE). (**e**) Key performance of different modules and die photo. (Reprinted from [61], Copyright 2009, with permission from IEEE).
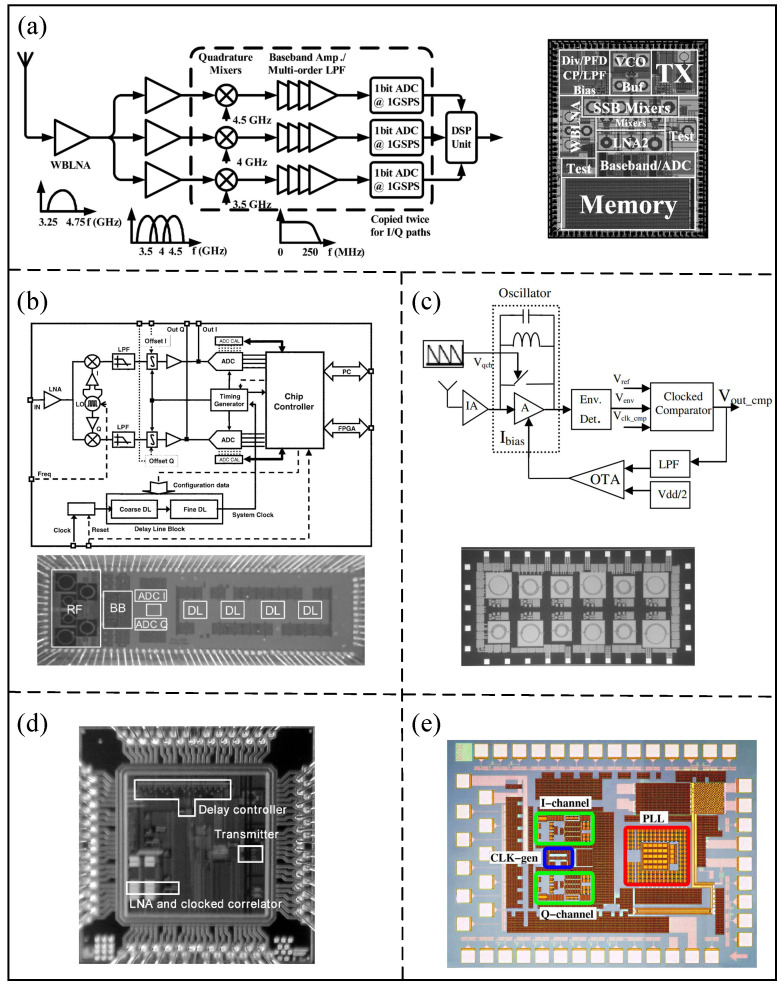



### 4.3. Comparison and Discussion

The parameters of UWB pulse radio receivers are compared in Table 3, including the fabrication process, bandwidth, maximum symbol rate, power consumption, and chip area. Obviously, the mainstream fabrication process of UWB pulse radio receivers is 0.18 μm CMOS technology. Compared with other receivers, the 0.13 μm process adopted by Hellepute [61] can improve the performance, and reduce power consumption and chip area. The architecture proposed by Anis [59] has a high bandwidth, which can transmit highspeed signals in an indoor situation. Due to the sub-1 GHz operating bandwidth, the signal transmitted by the architectures [60,61] is a low-speed signal with a strong penetrating ability and wide range. Compared with other receivers, the architecture proposed by Medi [57] achieves lower power consumption and a higher maximum bit rate, which makes it suitable for high-speed data communication.

## 5. Decoder

During the data transmission process, data can easily be disturbed by noise. The decoder corrects erroneous data by adding check bits during the encoding process. In the UWB transmitting link, the decoder module consists of a Viterbi decoder and an RS decoder.

### 5.1. Viterbi Decoder

The Viterbi algorithm was proposed in 1967 [62], and is used to produce a sequence of observed events. As a convolutional decoding method, the Viterbi decoding has been widely used in digital circuits and communication decoding. The Viterbi decoder consists of a branch measurement unit (BMU), an add comparison selection unit (ACSU), and a surviving path memory unit (SMU), as shown in Figure 10a. Recursive operation is a nonlinear feedback loop in the ACSU, and is a key method for improving the speed of the decoder [63].

In order to simplify the nonlinear feedback loop, Fettweis [64] proposed a semicircular algebraic architecture technique consisting of two recursive ACSU operations. The architecture changes the nonlinear recursion in the Viterbi algorithm to achieve linear recursion, which breaks the iteration bounds of the Viterbi decoding algorithm.

In order to reduce the length of the critical path, Parhi et al. [65] proposed a pipelined most significant bit (MSB) ACSU architecture. The length of the critical path is reduced to the iteration limit in the ACSU by balancing the establishment time of different paths, which can reduce the parallelism and area of Viterbi decoder. Compared with the traditional architecture, the critical path of ACSU can be reduced by 15% via the proposed architecture.

In addition, Goo et al. [66] proposed a pipelined MSB ACSU architecture based on the two-step look-ahead technique, as shown in Figure 10b. The two-step look-ahead technique is used to eliminate the feedback operation for improving the proposed decoder speed. Compared with a traditional MSB ACSU, the proposed decoder saves 12% of area and increases speed by 9%.

Kong et al. [67] proposed a low-latency branch precomputation architecture for a high-throughput Viterbi decoder, as shown in Figure 10c. The look-ahead technique is improved by changing the calculation of ACSU latency to increase logarithmically. The proposed architecture is used to avoid the linear stepwise calculation process and shows great improvement in reducing latency.

In order to reduce the latency of the M-step look-ahead method, Cheng et al. [68] proposed a Viterbi decoding method with a K-nested layer that can efficiently reduce latency by parallel work. The results show improved hardware efficiency and that the latency of ACSU calculation is reduced.

In the M-step look-ahead method, one step of the complex trellis consists of multiple steps of the chronological binary trellis, which is called branch metric precomputation (BMP). In order to simplify the BMP, Liu et al. [69] proposed an overall low-complexity BMP architecture based on a balanced binary grouping (BBG) algorithm, which can be used to eliminate redundancy and achieve minimum complexity and latency. The complexity/delay of the proposed architecture is reduced by 45.65%/72.50%.
Figure 10(**a**) Basic computation units in a Viterbi decoder. (Reprinted from [65], Copyright 2004, with permission from IEEE). (**b**) Block diagram of the proposed two−bit level pipelined MSB−first ACSU. (Reprinted from [66], Copyright 2008, with permission from IEEE). (**c**) An architecture of the low−latency branch precomputation Viterbi decoder. (Reprinted from [67], Copyright 2004, with permission from IEEE).
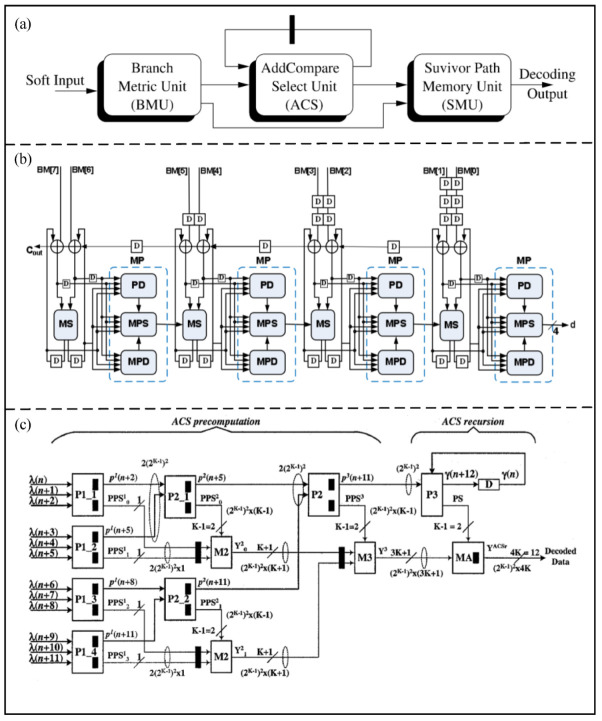



### 5.2. Comparison and Discussion of Viterbi Decoder

The latency and complexity of the ACSU for the lookahead architecture are summarized in Table 4, which represent the number of clock cycles and adders, respectively. In the table, K is the constraint length of the decoder and M is the step size of the look-ahead method. Compared with the conventional method, these proposed architectures are greatly optimized for latency by increasing the complexity. In the architecture proposed by Kong et al. [67], the K-layer look-ahead method is applied to the conventional architecture, which reduces the latency. The architecture proposed by Cheng et al. [68] reduces the complexity by improving the look-ahead method. In addition, the BBG method is applied to the proposed architecture by Kong et al. [67], which can significantly optimize latency and complexity. The architecture proposed by Liu et al. [69] performs best in terms of latency, and its hardware resources are slightly more complicated than those of a conventional architecture.

### 5.3. RS Decoder

An RS decoder is widely used to decode and correct data that have been encoded in various digital communication systems [21]. An RS decoder is divided into three parts: the calculation of correction factors, the solution of the key equation, and the determination of error location and size [70], as shown in Figure 11a. The solution of the key equation is the most complex part of the decoder, and can be implemented by the Berlekamp–Massey (BM) algorithm or the Euclidean algorithm.

#### 5.3.1. BM Algorithm

The BM algorithm was proposed by Berlekamp and Massey for solving the key equations in the decoder [71]. In the BM algorithm, the key equation is solved by decoding iteration, which leads to a complex operation.

In order to simplify the BM algorithm, Reed et al. [72] proposed an inverse-free decoding method, called iBM. The iBM algorithm eliminates the inversion operation of the BM algorithm in a nonbinary Galois domain. Therefore, a Very Large-Scale Integration (VLSI) of the RS code can be realized by the iBM algorithm. In addition, Sarwate et al. [73] proposed a new reconfigurable inverse-free decoding architecture called RiBM, as shown in Figure 11b. The critical path can be optimized by RiBM architecture, so the number of multipliers/adders is reduced from two/$1+\log_{2}\left(t+1\right)$ (where *t* is the number of detection errors) to one. Based on an RiBM architecture, Liang et al. [74] developed a compensated, simplified, inverse-free BM called CS-RIBM architecture. The redundant calculation in RiBM architecture can be eliminated by the CS-RiBM architecture, so high throughput and low hardware complexity can be achieved by the proposed architecture. Compared with the RiBM architecture, the proposed architecture can reduce the area by 14% and improve the efficiency by 17%. In order to further reduce the hardware complexity, Liu et al. [75] proposed a simplified inverse-free BM architecture with recursive compensation called RCS-RiBM. The proposed architecture consists of a processing element and a compensation unit, which can effectively reduce the hardware complexity. Compared with the common RS decoder, the proposed architecture can reduce the area by 11%. Based on the CS-RiBM architecture, Lu et al. [76] proposed a modified, compensated, simplified, inverse-free BM (MCS-RiBM) architecture, which combines the folding technology to improve the hardware utilization and simplify the generated architecture. Figure 11c shows the folded compensated simplified circuit diagram. The proposed architecture has a total gate number of 255,400.

In order to simplify the iBM architecture, Wu et al. [77] proposed an enhanced parallel inverse-free BM (EPIBM) architecture, as shown in Figure 11d. The generalized Horiguchi–Koetter formula is used in the proposed architecture to increase its parallel calculation ability. Compared with the iBM architecture, which requires 3*t* systolic cells, the EPIBMA architecture only needs 2*t* + 1 systolic cells. In addition, Ji et al. [78] proposed the recursive enhanced parallel inverse-free BM (REPIBM) architecture based on the 0.18 um CMOS process, which can effectively reduce the hardware complexity by recursive operation. The experimental result shows the number of gates is 13,000.
Figure 11(**a**) Block diagram of RS decoder. (Reprinted from [75], Copyright 2017, with permission from IEEE). (**b**) The systolic RiBM architecture. (Reprinted from [73], Copyright 2001, with permission from IEEE). (**c**) The folded compensated simplified circuit diagram. (Reprinted from [76], Copyright 2019, with permission from IEEE). (**d**) EPIBM block diagram. (Reprinted from [77], Copyright 2015, with permission from IEEE). (**e**) The process of fractional folding. (Reprinted from [79], Copyright 2021, with permission from IEEE).
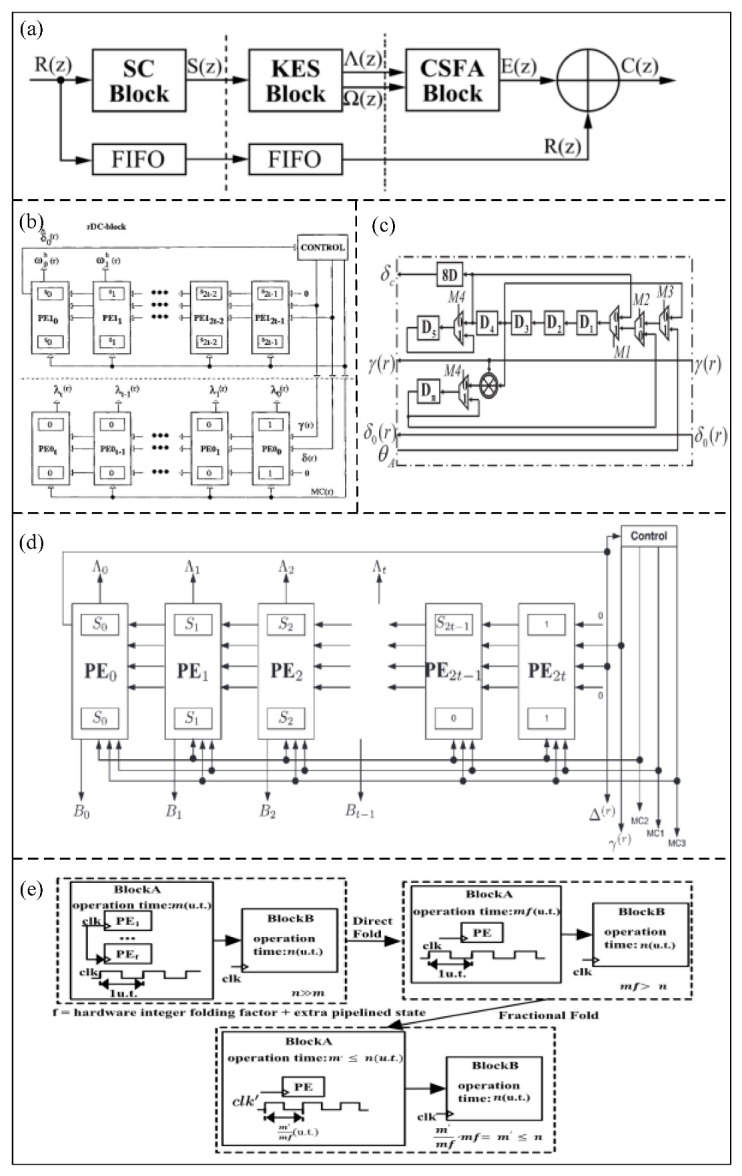



Liu et al. [79] proposed an enhanced parallel inverse-free BM RS decoder based on fractional folding (FF-EPIBM). The process of fractional folding is shown in Figure 11e. The number of processing units can be reduced to one, resulting in ultra-low hardware complexity. Compared with the fully extended parallel EPIBM architecture, the RS decoder based on the FF-EPIBM architecture can reduce the hardware complexity by about 60%.

#### 5.3.2. Euclidean Algorithm

The Euclidean algorithm is also a common method used to solve the key equations in the RS decoder, which can be achieved by calculating the greatest common factor. The complexity of the Euclidean algorithm is high due to its inversion operation. To eliminate the inversion operation, Shao et al. [80] proposed a modified Euclidean (ME) architecture to calculate the error location polynomial. In the ME architecture, the degree calculation can increase the complexity of circuit. Bae et al. [81] proposed a modified Euclidean architecture to remove the degree calculation (DCME), as shown in Figure 12a. The proposed architecture has low hardware complexity because of completely removing the degree calculation. Compared with the conventional ME decoder, the proposed DCME architecture can reduce the total gate count and delay by 23% and 10%, respectively. In order to increase speed and reduce complexity, Lee et al. [82] proposed a pipelined DCME (PDCME) architecture for the RS decoder, as shown in Figure 12b. The proposed architecture can reduce the total gate count by 15% more than the ME architecture.

In the DCME architecture, the conventional systolic architecture needs many processing elements, which can increase the circuit complexity. Yuan et al. [83] proposed a high-speed and low-complexity Reed–Solomon (RS) decoder architecture based on a recursive DCME architecture, as shown in Figure 12c. The proposed architecture uses the recursive architecture of a single processing element instead of the conventional architecture of multiple processing elements, which has low hardware complexity. The proposed architecture can reduce the total gate count by 30% compared with the DCME architecture.

In order to replace the ME architecture, Baek et al. [84] proposed a simplified Euclidean (SE) decoder architecture, as shown in Figure 12d. In the proposed architecture, the new initial conditions and polynomials are used, which can significantly reduce the complexity. The total gate count of the proposed architecture is only 40,136 for the (255,239) RS code, and is reduced by 5% compared with the DCME architecture. In addition, Hsu et al. [85] proposed an RS decoder architecture based on the real-time folding modified Euclidean algorithm, as shown in Figure 12e. Compared with the parallel RS architecture, the proposed architecture can reduce the hardware complexity by about 50%.
Figure 12(**a**) Proposed DCME architecture. (Reprinted from [81], Copyright 2006, with permission from IEEE). (**b**) Block diagram of DCME algorithm. (Reprinted from [82], Copyright 2007, with permission from IEEE). (**c**) Block diagram of RDCME algorithm. (Reprinted from [83], Copyright 2009, with permission from IEEE). (**d**) New SE algorithm. (Reprinted from [84], Copyright 2013, with permission from IEEE). (**e**) Proposed Jit−FMEA algorithm. (Reprinted from [85], Copyright 2004, with permission from IEEE).
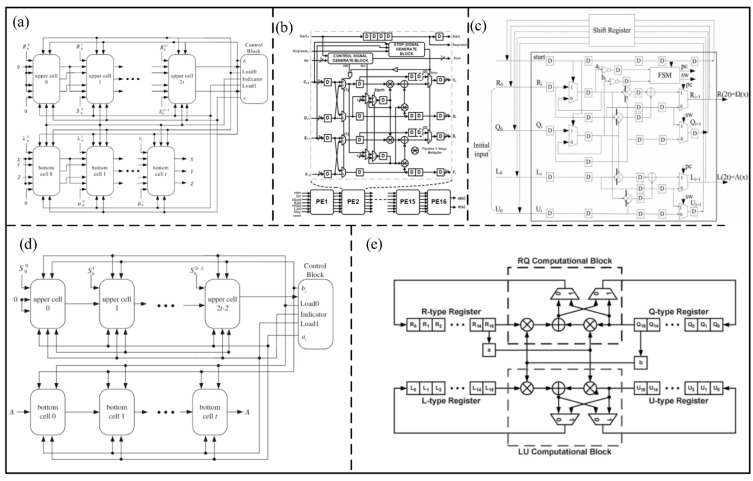



### 5.4. Comparison and Discussion for the RS Decoder

The performance parameters of different RS decoders are compared in Table 5, and mainly included technology, total gates, frequency, throughput, and technology-scaled normalized throughput (TSNT). Obviously, the mainstream fabrication process of the RS decoders is the 0.18 μm CMOS technology. However, advanced processes with smaller feature sizes, such as 0.13, 0.09, and 0.04 μm CMOS technologies, have been adopted since they can greatly increase the working frequency and throughput. The total number of gates of the RS decoder consists of the syndrome computation (SC), the Chien search and error evaluation (CSEE) and the key equation solver (KES) blocks. Compared with other architectures, only one processing unit is used in the folding architecture proposed by Liu et al. [79], which can effectively reduce the number of gates used in the KES. In addition, the throughput and TSNT of the proposed architecture are 24.8 Gb/s and 592 Mb/s/K·gate due to processing data in parallel and high frequency. The recursive method is used in the RCS-RiBM architecture [75] and RDCME [83] to reduce the total number of gates to fewer than 20,000. When the frequency is less than 700 MHz, the architecture proposed by Lu et al. [76] achieves the highest throughput, but the compensation unit used in the architecture leads to an increase in the number of selectors and path delay.

## 6. Conclusions and Outlook

The development of the UWB physical layer, including the encoder, pulse generator, receiver, and decoder, is introduced in this paper. The encoder includes an RS encoder, a convolutional encoder, and a scrambler. In the RS encoder, methods to increase coding speed and reduce hardware resources are discussed. The PSD can be reduced by a convolutional encoder and scrambler. In the pulse generator, the low spectral interference generator can meet IEEE 802.15.4a, and the optional channel generator and low power consumption generator should meet the spectrum requirements. In the receiver, the rake receiver is a key technology of spread-spectrum communication systems, and the analog front-end receiver can amplify and digitize the received signal. In the decoder, the delay of the Viterbi decoder can be reduced by optimizing algorithms. The RS decoder is optimized by BM and Euclidean algorithms to achieve higher speed and lower hardware complexity. A detailed discussion of the UWB physical layer is presented to provide suggestions for the design of a high-performance UWB physical layer.

With the requirements of high speed, low power, and fewer hardware resources, the performance of a UWB system can be improved in several aspects:The encoder speed should be improved.

The encoder can be blocked when encoding large amounts of data, and the coding efficiency can be reduced. Besides the traditional bit-serial RS encoder, other architectures such as the bit-parallel and dual-based encoders can be adopted to improve the encoder speed. However, these architectures can influence the area and power consumption, so the power consumption and area should be balanced when the encoder speed is increased.

The transmitting link should meet the low PSD requirement.

UWB signals easily interfere with narrowband signals because of their wide frequency spectrum. Therefore, the UWB frequency should meet the low PSD requirement. The MFD algorithm in a convolutional coder, two-layer LFSR of a scrambler, and on-chip balun in a pulse generator can effectively reduce the power spectral density.

The pulse generator should maintain low power consumption.

UWB is widely used in wireless wearable devices, so low power consumption is required to support long-term usage. Existing methods include reducing the peak current of the pulse generator and maintaining the off state when the pulse generator is out of work. More methods are needed to maintain the low power consumption of the pulse generator.

Multiple-user interference can be decreased in the receiver.

Multiple-user access and communication exist when UWB is used for indoor positioning, so the effect of multiple-user interference and environmental noise should be decreased by orthogonal frequency division multiplexing to improve the anti-multipath interference and robustness.

The complexity and latency of decoders should be reduced.

Due to the large volume of resources used by the decoders, the BBG algorithm can be used in different Viterbi decoders to reduce the complexity and latency of BMP units. In order to reduce the complexity of the key equation in the RS decoder, the generalized fractional folding and pipeline structures can be used in different algorithms.

## Figures and Tables

**Figure 1 micromachines-14-00008-f001:**
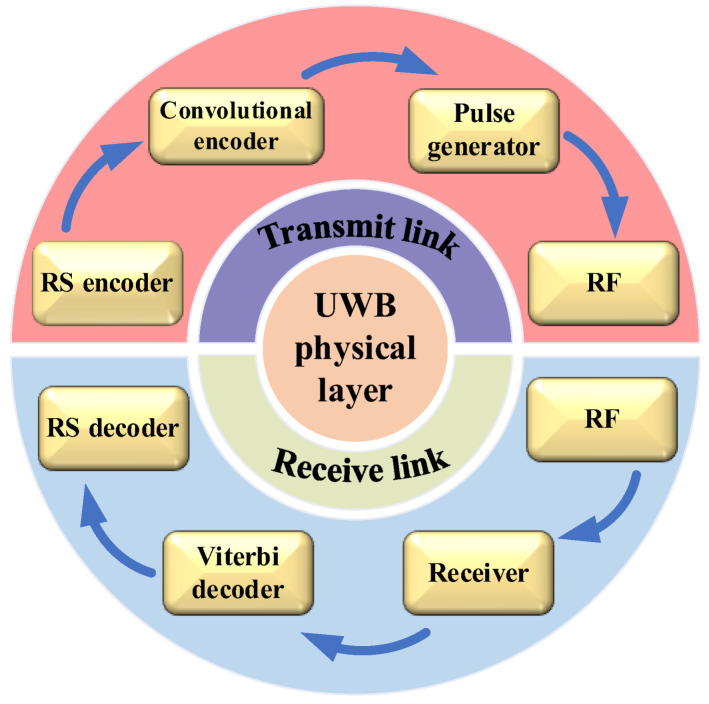
Structure of UWB physical layer.

**Figure 3 micromachines-14-00008-f003:**
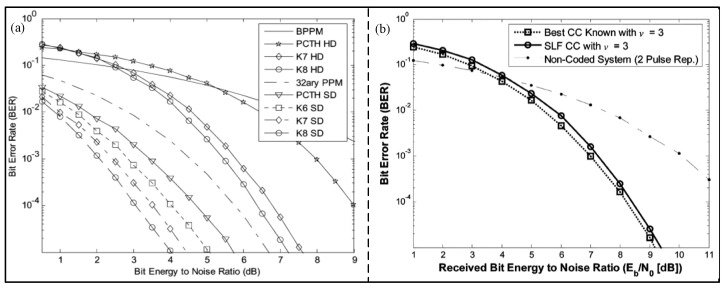
(**a**) BER performance for several of the rate 1 binary to 32−ary encoders and 32−ary PCH. (Reprinted from [32], Copyright 2007, with permission from IEEE). (**b**) BER performance in IEEE 802.15.4a CM1. (Reprinted from [33], Copyright 2011, with permission from IEEE).

**Figure 4 micromachines-14-00008-f004:**
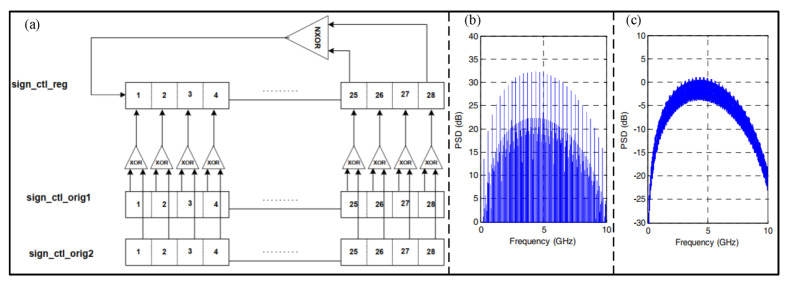
(**a**) Architecture of the two−layer LFSR. (**b**) PSD of original scrambler. (**c**) PSD of proposed scrambler. (Reprinted from [35], Copyright 2004, with permission from IEEE).

**Figure 5 micromachines-14-00008-f005:**
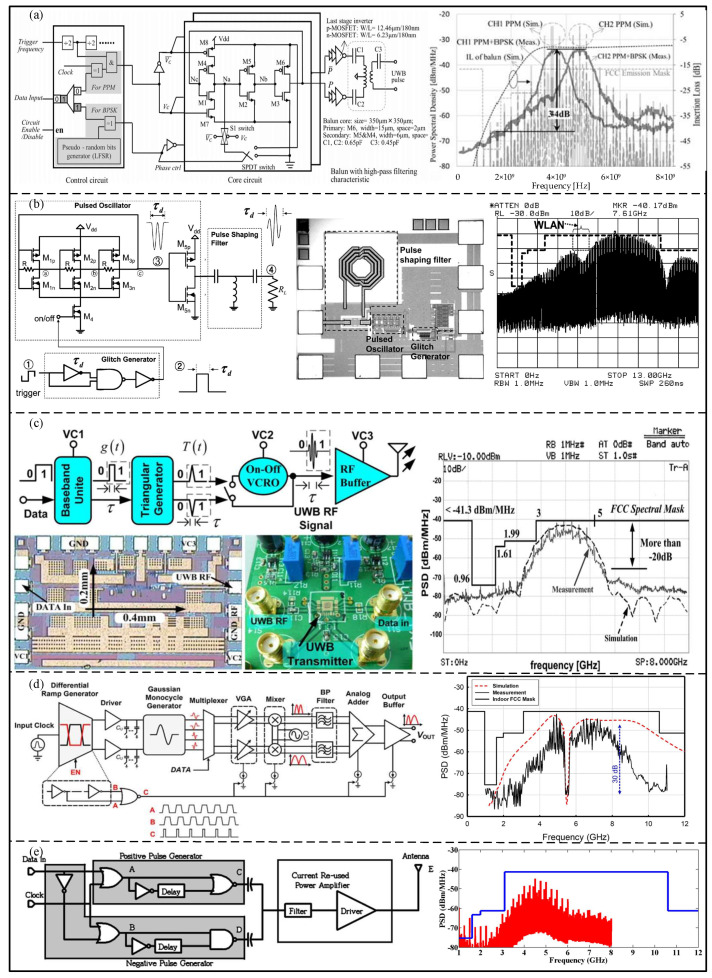
(**a**) Block diagram and output PSD of proposed UWB pulse generator. (Reprinted from [39], Copyright 2017, with permission from IEEE). (**b**) Schematic, die photograph of the proposed UWB pulse generator. (Reprinted from [37], Copyright 2009, with permission from IEEE). (**c**) Block diagram, die photograph, and test board of the OOK pulse generator, and output PSD in compliance with FCC mask. (Reprinted from [38], Copyright 2013, with permission from IEEE). (**d**) Block diagram of UWB pulse generator, simulated and measured spectrum of UWB pulse. (Reprinted from [40], Copyright 2013, with permission from IEEE). (**e**) Block diagram and Measured PSD of the IR−UWB BPSK transmitter. (Reprinted from [41], Copyright 2017, with permission from IEEE).

**Figure 6 micromachines-14-00008-f006:**
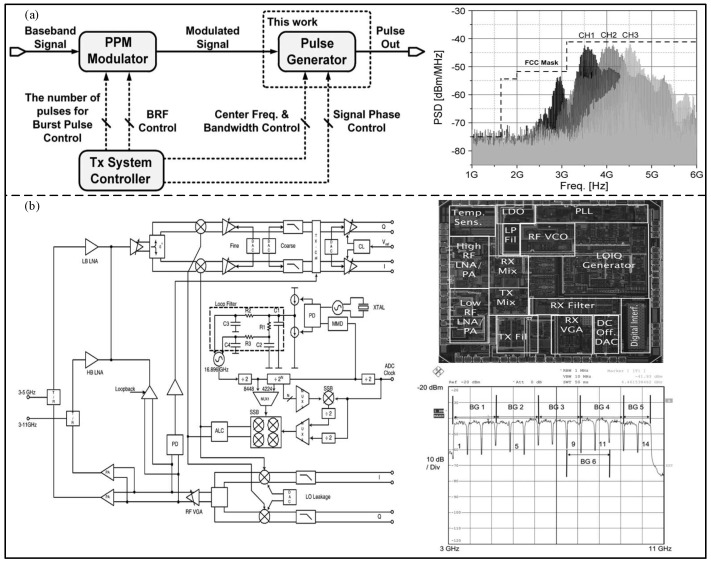
(**a**) System configuration for UWB transmission and measured output for each channel. (Reprinted from [42], Copyright 2012, with permission from IEEE). (**b**) Block diagram and chip microphotograph of the UWB transceiver, and the measured spectrum of all 14 channels. (Reprinted from [44], Copyright 2007, with permission from IEEE).

**Figure 7 micromachines-14-00008-f007:**
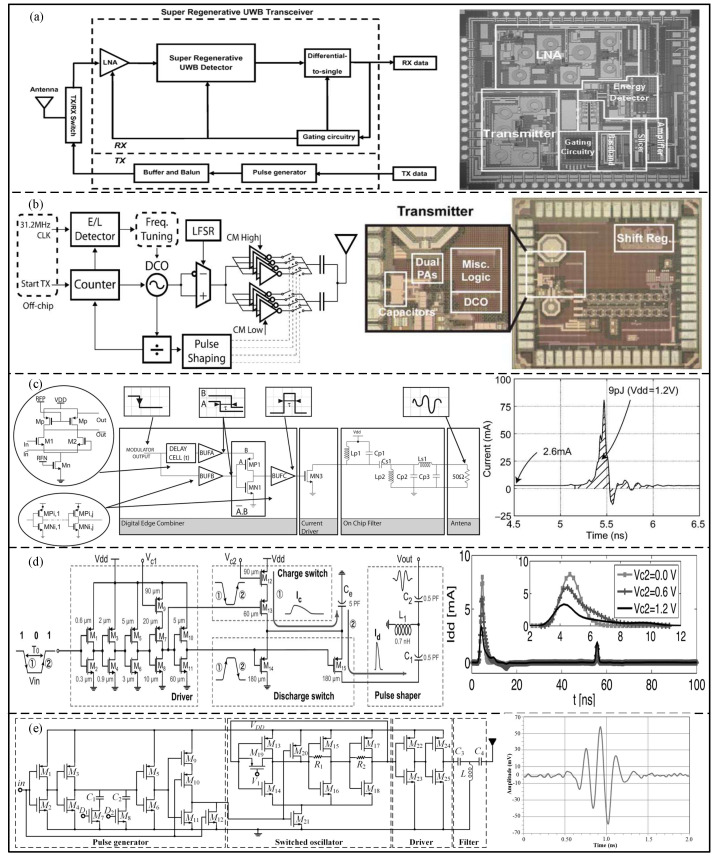
(**a**) Block diagram and die photograph of the super−regenerative UWB transceiver. (Reprinted from [46], Copyright 2014, with permission from IEEE). (**b**) Block diagram and die photograph of the proposed generator. (Reprinted from [47], Copyright 2009, with permission from IEEE). (**c**) Architecture of the proposed pulse generator and current consumption simulation. (Reprinted from [48], Copyright 2010, with permission from IEEE). (**d**) CMOS implementation of the pro-posed pulse generator and measured current at a PRR of 10 Mpps. (Reprinted from [49], Copyright 2014, with permission from IEEE). (**e**) Detailed schema and measured output waveform of the proposed transmitter. (Reprinted from [50], Copyright 2020, with permission from IEEE).

**Table 1 micromachines-14-00008-t001:** Comparison between different encoders.

Modulation	Style	Scope	Effect
RS encoder	Forward error correction code	PSDU	Reduce BER
Convolutional code	Forward error correction code	PSDU + PHR	
Scrambler	-	Preamble code	

**Table 2 micromachines-14-00008-t002:** Summary of UWB pulse generators in different architectures.

Ref.	Tech. (μm)	Freq. (Ghz)	BW (Ghz)	Number of Channels	EP (pJ/Pause)	Modulation	Architecture	Area (mm^2^)	Vpp (V)	PRR (Mpps)
[39]	CMOS 0.18	3–5	1.2	2	32	PPM + BPSK	Pulsed oscillator	-	0.22	125
[37]	CMOS 0.18	6–10	4.5	1	27.6	OOK	Pulsed oscillator	0.11	0.673	
[38]	CMOS 0.18	3–5	2	1	20	OOK	Pulsed oscillator	0.08	0.26	250
[40]	CMOS 0.09	-	5.5	-	65	BPSK	Analog	1.9	-	400
[41]	CMOS 0.18	3.5–6.5	-	-	86	BPSK − PAM	-	0.22	1.8	
[42]	CMOS 0.13	3.1–4.8	0.5	3	48	PPM + BPSK	Delay lines	-	0.42	50
[43]	CMOS 0.065	3.1–4.8	0.5	3	30	PPM + BPSK	Delay lines	0.182	0.22	200
[44]	SiGe BiCMOS 0.13	3.1–10.6	0.5	14	791	-	Analog	25	-	480
[45,46]	CMOS 0.18	3–5	1.36	-	671	OOK	-	4.4	2	-
[47]	CMOS 0.09	3–5	<0.5	3	17.5	PPM + BPSK	-	0.07	0.71	-
[48]	CMOS 0.13	3.1–10.6	6.8	-	9	OOK	Filtered Edge Combination	0.54	1.42	-
[49]	CMOS 0.18	3–10	7.5 6.75 6	-	18 14.6 5	OOK	-	0.16	-	10 250 1000
[50]	CMOS 0.18	3–8.8	4.75	-	3	OOK	-	0.35	0.13	

**Table 3 micromachines-14-00008-t003:** Parameters of UWB pulse radio receiver.

Ref.	Technology	Band (GHz)	Pulses/s	Energy Per Pulse	Chip Area
Medi et al. [57]	0.18 μm	3–5	1 G	98 pJ	15.05 mm^2^
Ryckaert et al. [58]	0.18 μm	3–5	20 M	1.44 nJ	-
Anis et al. [59]	0.18 μm	6–8.5	8 M	250 pJ	3.15 mm^2^
Terada et al. [60]	0.18 μm	sub-1	1 M	300 pJ	0.38 mm^2^
Helleputte et al. [61]	0.13 μm	sub-1	39.0625 M	70 pJ	1.5 mm^2^

**Table 4 micromachines-14-00008-t004:** Latency and complexity of the ACSU.

M	4	8	12	16	20	24	28	32	36
Latency (K = 4)									
conventional	4	12	20	28	36	44	52	60	68
Kong et al. [67]	4	8	12	12	16	16	16	16	20
Cheng et al. [68]	4	8	12	16	14	16	18	20	20
Kong et al. (BBG) [69]	2	6	10	10	14	14	14	14	18
Liu et al. [69]	2	6	10	10	14	14	14	14	18
Complexity (K = 4)									
conventional	288	1056	1824	2592	3360	4128	4896	5664	6432
Kong et al. [67]	288	1536	2784	4032	5280	6528	7776	9024	10,272
Cheng et al. [68]	288	1536	2304	3072	4800	5568	6336	7104	8832
Kong et al. (BBG) [69]	256	1472	2688	3904	5120	6336	7552	8768	9984
Liu et al. [69]	256	1472	2688	3904	4672	5440	6208	8768	9536

**Table 5 micromachines-14-00008-t005:** Performance summary of recently proposed RS decoders for the UWB physical layer.

Architecture	Tech. (μm)	SC and CSEE	KES	Total Gates	$f_{max}$ MHz	Throughput (Gb/s)	TSNT (Mb/s/K·Gate)
Liang et al. [74]	0.18	6280	15,340	21,620	370	2.96	189.57
Liu et al. [75]	0.18	7000	5963	12,963	640	5.1	544.75
Lu et al. [76]	0.09	14,650	10,890	25,540	625	16.2	439.1
Liu et al. [79]	0.04	7993	4896	37,554	3100	24.8	592.0
Baek et al. [81]	0.25	20,453	21,760	42,213	200	-	-
Lee et al. [82]	0.13	7000	46,200	53,200	660	5.3	-
Yuan et al. [83]	0.18	7000	11,400	18,400	640	5.1	383.78
Baek et al. [84]	0.18	22,336	17,800	40,136	370	2.96	-
Hsu et al. [85]	0.18	-	-	20,614	400	3.2	214.62

## Data Availability

Not applicable.

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
