# Peer review of "The Development and Progress of the UWB Physical Layer"

_micromachines, 2022, doi:10.3390/mi14010008_

Round 1

Reviewer 1 Report (Previous Reviewer 3)

This paper reviewed the recent development of UWB physical layer, including encoders, pulse generators, receivers, and decoders. Detailed comparison and discussion of different components of the UWB physical layer were presented to provide suggestions for the design of a high-performance UWB physical layer.

This version of the manuscript seems better than the previous one in terms of contents and expressions. However, there are still several typos that the authors should take care of before being considered for publication. These typos are marked in red as follows.

1.       In the UWB physical layer, the transmitting link, including an encoder and a pulse generator, is used to improve the anti-interference ability of signal. While the receiving link, including a receiver and a decoder, can correct the error signal.

2.       the UWB  29 system has been widely applied in many fields such as indoor positioning, [4-8] wireless body area network [9-12] and, radar, [13-15].

3.       In this paper, the development and prospects of UWB physical layer transceiver are reviewed to provide valuable references for designing high performance of UWB physical  layer.

4.       [41] proposed a low power, high peak value UWB pulse generator based on the 0.18 μm process for 6-10 GHz, which is consisted of a pulse generator, a pulse shaping filter and a pulse oscillator, as shown in Figure 5. (b).

5.       Which is consisted of a pulse generator, a pulse shaping filter and a pulse oscillator, as shown in Figure 5. (b). The pulse shaping filter was used to make the spectrum comply with Federal Communications Commission (FCC) regulations. For the GPS band in 0.96-1.61 GHz, the proposed pulse shaping filter can suppress the spectral component by more than 34 dB.  As shown in Figure 5 (c),

Author Response

Response: Thanks for the reviewer’s suggestive comments. The manuscript has been carefully checked again, and the English language has undergone English language editing by MDPI. The English editing ID is english-56273. In the revised manuscript, some phrases and sentences have been modified. Thanks!

The modifications are listed as follows:

  1. In the UWB physical layer, the transmitting link, including an encoder and a pulse generator, is used to improve the anti-interference ability of the signal, while the receiving link, including a receiver and a decoder, can correct the error signal.
  2. Due to the advantages of low power, small size, and strong penetration, the UWB system has been widely applied in many fields such as indoor positioning [4-8], wireless body area network [9-12], and radar [13-15].
  3. In this paper, the development and prospects of UWB physical layer transceiver are reviewed to provide valuable references for designing a high-performance UWB physical layer.
  4. Additionally, Sim et al. [41] proposed a low-power, high-peak-value UWB pulse generator based on the 0.18 μm process for 6–10 GHz, which consisted of a pulse generator, a pulse-shaping filter, and a pulse oscillator, as shown in Figure 5b.
  5. The pulse-shaping filter was used to make the spectrum comply with Federal Communications Commission (FCC) regulations. For the GPS band at 0.96–1.61 GHz, the proposed pulse-shaping filter can suppress the spectral component by more than 34 dB. As shown in Figure 5c, Zhao et al. [42] proposed a CMOS UWB pulse generator for 3–5 GHz with on–off keying (OOK) modulation, in which a new push-and-pull-integrating narrow tri-angular pulse generator was designed to reduce the common mode interference and static current.

Reviewer 2 Report (New Reviewer)

In the manuscript “The development and progress of the UWB physical layer” submitted for publication in the journal Micromachines, the authors reviewed and compared the structure and performance of the codec and transceiver of the UWB physical layer. It also covers some typical architectures and features which provide valuable references and suggestions for the design of UWB physical layer. This manuscript reviewed most of the recent important findings and provided the development direction for future UWB design such as high speed, low power consumption and less hardware resource. The manuscript is well prepared and scientifically sound, therefore I support this manuscript to be published.

Author Response

Response: Thank you for your recognition.

This manuscript is a resubmission of an earlier submission. The following is a list of the peer review reports and author responses from that submission.

Round 1

Reviewer 1 Report

The title of the article corresponds to its content and contains the main theses that are correlated with the text title. The manuscript covers a relevant research topic. The research topic is relevant because Ultra-Wideband is still a very promising technology with the possibility of further development. The article outlines the methodology used in a proper manner. The article has been written with solid scientific language.
Critical remarks:

-       The abstract is very general and should be enhanced.

-       The final part of the article (6. Conclusion and outlook) seems to be unrelated to the content of the work. There are no references to individual parts of the article (e.g. subsection 2.2).

-       The figures are overloaded with content. It should be considered whether all the information they contain is necessary.

Author Response

Dear reviewer and editor,

Thanks for the suggestive advice from reviewers and editor. The manuscript has been carefully checked again, and the English language has been improved. The followings are the response to the comments from the reviewers and editor. The corresponding modifications have been made and the important corrections are highlighted in YELLOW in the revised manuscript.

The title of the article corresponds to its content and contains the main theses that are correlated with the text title. The manuscript covers a relevant research topic. The research topic is relevant because Ultra-Wideband is still a very promising technology with the possibility of further development. The article outlines the methodology used in a proper manner. The article has been written with solid scientific language.

Q1. The abstract is very general and should be enhanced.

Response: Thanks for the reviewer’s suggestive comments. The abstract has been enhanced. First, the influence of UWB physical layer on the speed and quality of signal transmission is emphasized. In addition, the structure of the transceiver link has been supplied, which includes an encoder, a pulse generator, a receiver and a decoder. Finally, the function of transceiver link is described. The transmit link is used to improve the anti-interference ability of signal, and the error signal can be corrected by the receive link. The corresponding modifications have been made in ‘Abstract’. Thanks!

Q2. The final part of the article (6. Conclusion and outlook) seems to be unrelated to the content of the work. There are no references to individual parts of the article (e.g. subsection 2.2).

Response: Thanks for the reviewer’s suggestive comments. In Section 6, the paper has been summarized, which is related to the content of the work. The modifications of Section 6 are as follows. ‘The encoder includes RS encoder, convolutional encoder and scrambler. In RS encoder, the methods to increase coding speed and reduce hardware resources are discussed. The PSD can be reduced by convolutional encoder and scrambler. In pulse generator, low spectral interference generator can make the spectral meet IEEE 802.15.4a, and the optional channel generator and low power consumption generator should meet the spectrum requirements. In receiver, rake receiver is a key technology of spread spectrum communication system, and the analog front end receiver can amplify and digitize the received signal. In decoder, the delay of Viterbi decoder can be reduced by optimizing algorithms. The RS decoder is optimized by BM and Euclidean algorithms to achieve higher speed and lower hardware complexity. The detailed comparison and discussion of UWB physical layer are presented to provide suggestions for the design of high-performance UWB physical layer.’

In addition, Section 6 provides suggestions for the future development tendency, including the improvement of the encoder speed, low PSD requirement of the transmitting link, low power consumption of the pulse generator, low multiple-user interference, low complexity and latency of decoders. Low PSD requirement of the transmitting link is added. The PSD can be reduced by the MFD algorithm in the convolutional encoder, the two-layer LFSR of scrambler and the on-chip balun in the pulse generator. The corresponding modifications have been made in SECTION6.Conclusion and outlook’. Thanks!

Q3. The figures are overloaded with content. It should be considered whether all the information they contain is necessary.

Response: Thanks for the reviewer’s suggestive comments. The figures have been simplified and modified. Some unnecessary pictures have been removed. The modifications have been made in Figures 2, 9 and 11. Thanks!

Reviewer 2 Report

The paper provides an overview of recent (2004 an onwards when leaving out a few classical references) developments of the UWB physical layer and in doing so the paper aims to outline the progress of the same physical layer.

The reference list seems reasonable albeit not exhaustive. Also, out of some 80+ references only around 30 are 10 years or less old and for a paper that aims to outline progress this seems strange. 

A key weakness of the work is that the developments and progress is reported by more or less mentioning pieces of work from the public literature. No trends and tendencies, neither implementation nor performance wise, is given to any substantial degree. The work therefore appears unstructured and somewhat random and as a reader it is hard to determine what the "take away message is", if any. The first few pages contains a lot of basic textbook stuff that clearly should be omitted. Table 1 brings no value and almost every single figure is overcrowded, messy and hard to read/see. Layout pictures are included but they provide no value to the work and therefore only serves to add to the mess.  Copyright issues may also be relevant to consider, as most figures are simply copy'n'pasted from the original works.

The language needs to be updated. In a few cases sentences even do not make sense. Quite many typos are also found throughout the paper.

While I do recognize the significant work that has gone into this, then I cannot recommend the paper for publication in its current form. In my view a complete and significant reworking of the paper is needed, to rectify language and presentation, but more importantly to demonstrate tends, if any. In its current form there is no value for the reader.

Author Response

Dear reviewer and editor,

Thanks for the suggestive advice from reviewers and editor. The manuscript has been carefully checked again, and the English language has been improved. The followings are the response to the comments from the reviewers and editor. The corresponding modifications have been made and the important corrections are highlighted in YELLOW in the revised manuscript.

 The paper provides an overview of recent (2004 an onwards when leaving out a few classical references) developments of the UWB physical layer and in doing so the paper aims to outline the progress of the same physical layer.

Q1. The reference list seems reasonable albeit not exhaustive. Also, out of some 80+ references only around 30 are 10 years or less old and for a paper that aims to outline progress this seems strange.

Response: Thanks for the reviewer’s suggestive comments. This paper introduces the development and research of UWB physical layer during the last several decades. Due to the large time span of the references, the title has been changed as ‘The development and progress of UWB physical layer’. In addition, some references have been added to enrich the content. The improvements are as follows:

In the introduction, Refs. [1], [2], [3] introduces the applications of UWB in indoor positioning, wireless body area network and radar. In Section 3.1, Ref. [4] proposes a 250 Mb/s data rate IR-UWB transmitter, which is composed of a pulse generator and a current re-used power amplifier. Its peak PSD power is -42 dBm. In Section 3.3, Ref. [5] proposes a low power IR-UWB transmitter, consisting of a controllable pulse generator, a switchable tunable oscillator, a driver and a pulse shaping filter. Its power consumption is 3pJ/pause. In Section 5.3.1, Ref. [6] proposes the recursive enhanced parallel inverse-free BM (EPIBM) architecture to reduce hardware complexity. The detailed modifications have been made highlighted in YELLOW in the revised manuscript. Thanks!

  1. E. Puschita et al. Performance Evaluation of the UWB-based CDS Indoor Positioning Solution. International Workshop on Antenna Technology (iWAT). 2020; pp. 1-4.
  2. P. Promsrisawat and S. Promwong. A study of HB-UWB transfer function model for wireless body area network. International Conference on Digital Arts, Media and Technology (ICDAMT). 2018; pp. 225-228.
  3. S. Chen and H. H. Liu. UWB slot antenna on shielding can for high accuracy positioning application. Global Congress on Electrical Engineering (GC-ElecEng). 2020; pp. 43-45.
  4. P. Gunturi, N. W. Emanetoglu and D. E. Kotecki. A 250-Mb/s Data Rate IR-UWB Transmitter Using Current-Reused Technique. in IEEE Transactions on Microwave Theory and Techniques. Nov. 2017; vol. 65, no. 11, pp. 4255-4265.
  5. J. Radic, M. Brkic, A. Djugova, M. Videnovic-Misic, B. Goll and H. Zimmermann. Area and Power Efficient 3–8.8-GHz IR-UWB Transmitter With Spectrum Tunability. in IEEE Microwave and Wireless Components Letters. Jan. 2020; vol. 30, no. 1, pp. 39-42.
  6. W. Ji, W. Zhang, X. Peng and Y. Liu, High-efficient Reed Solomon decoder design using recursive Berlekamp-Massey architecture. IET Communications. March. 2016; vol. 10, no. 4, pp. 381-386.

Q2. A key weakness of the work is that the developments and progress is reported by more or less mentioning pieces of work from the public literature. No trends and tendencies, neither implementation nor performance wise, is given to any substantial degree. The work therefore appears unstructured and somewhat random and as a reader it is hard to determine what the "take away message is", if any.

Response: Thanks for the reviewer’s suggestive comments. Based on the IEEE 802.15.4a UWB physical layer transceiver link, the architecture of this paper is elaborated, which includes encoder, pulse generator, receiver and decoder. In this review, the content is arranged according to the architecture of UWB physical layer. The development and progress of UWB physical layer are introduced with the improvement of performance, such as coding speed, hardware complexity, power consumption and PSD. In addition, the development of fabrication process is considered in the review of UWB physical layer. Finally, the development trend of UWB physical layer is presented in conclusion and outlook. Thanks!

Q3. The first few pages contain a lot of basic textbook stuff that clearly should be omitted.

Response: Thanks for the reviewer’s suggestive comments. In the revised manuscript, the basic textbook stuff has been deleted or omitted. Especially, the composition of encoder and decoder has been deleted. The corresponding modifications have been made in Introduction. Thanks!

Q4. Table 1 brings no value and almost every single figure is overcrowded, messy and hard to read/see.

Response: Thanks for the reviewer’s suggestive comments. The number of XOR gates of Galois domain multiplier in RS encoder can be reduced by subexpression sharing and logic algebra algorithm. The number of XOR gates in the multiplier with two different algorithms is compared in Table I. Due to overcrowded figure in Table I, it is deleted and replaced by ‘For RS (255,239) encoder, the number of total XOR gates without optimization algorithm is 366, however, those in [26] and [27] are 276 and 246, respectively.’. The corresponding modification has been made in SECTION2.1. Rs encoder’. Thanks!

Q5. Layout pictures are included but they provide no value to the work and therefore only serves to add to the mess.  

Response: Thanks for the reviewer’s suggestive comments. The figures have been simplified and modified. Some valueless pictures have been removed. The modifications have been made in Figures 2, 9 and 11. Thanks!

Q6. Copyright issues may also be relevant to consider, as most figures are simply copy'n'pasted from the original works.

Response: Thanks for the reviewer’s suggestive comments. The copyright of the cited figures has been permitted by the publisher and all images are annotated. Thanks!

Q7. The language needs to be updated. In a few cases sentences even do not make sense. Quite many typos are also found throughout the paper.

Response: Thanks for the reviewer’s suggestive comments. The manuscript has been carefully checked again, and the English language has been improved. In the revised manuscript, some phrases and sentences have been modified. Thanks!

The modifications are listed as follows:

  1. Finally, the outlook of UWB physical layer is presented, and its development direction mainly includes high speed, low power consumption and less hardware resource.
  2. Due to the advantages of low power, small size, and strong penetration, the UWB system has been widely applied in many fields such as indoor positioning, [4-5,6,7,8 ] wireless body area network [9-10,11,12 ] and, radar, [13-14,15] etc.
  3. Therefore, the investigation of UWB physical layer is essential.
  4. The UWB physical layer transceiver link structure is as shown in Figure 1 [16]. RS encoders are used for payload bit from the physical layer to increase the anti-jamming ability of data [17].
  5. The anti-multipath capability of the receiver should be improved;
  6. The encoder is used to add a check bit for data, which can improve its anti-interference ability in channel transmission.
  7. In order to deal with this problem, Ren et al. [27] proposed a bit parallel multiplication RS encoder based on the dual basis, as shown in Figure 2 (a).
  8. In the last years, pseudo-chaotic jump time (PCTH) technique has been used in convolutional coders to suppress spectral lines and reduce bit error rate (BER).
  9. Mo et al. [36] proposed a scrambler with a two-layer LFSR to increase the randomness of the scrambler.
  10. In addition, Kouassi et al. [37] proposed a UWB transmitter with a random pulse-width scrambling method.
  11. A pulse generator is used to load encoded data onto the carrier, which is essential in UWB physical layer.
  12. The UWB system may conflict with other narrowband systems, so many circuits have been proposed to reduce PSD [38,39].
  13. The on-chip balun with high-pass characteristics has high suppression to PSD in the Global Positioning System (GPS) band.
  14. Also, Sim et al. [41] proposed a low power, high peak value UWB pulse generator based on the 0.18 μm process for 6-10 GHz, which is consisted of a pulse generator, a pulse shaping filter and a pulse oscillator, as shown in Figure 5. (b).
  15. In addition, a new on-off voltage-controlled ring oscillator (VCRO) with a complementary switch mode was also proposed, which can reduce power consumption by avoiding generating base-band energy.
  16. Based on the 90 nm CMOS process, Hedayati et al. [43] proposed a fully integrated analog UWB pulse transmitter with BPSK modulation, as shown in Figure 5. (d), which can coexist with IEEE 802.11a system.
  17. In order to solve this problem, Choi et al. [45] proposed an all-digital pulse generator based on the 0.13 μm CMOS technology, consisting of a delay line structure, as shown in Figure 6 (a).
  18. Based on the 65 nm CMOS process, Na et al. [46] proposed an all-digital UWB pulse generator with three optional channels.
  19. As shown in Figure 6 (b), the proposed pulse generator includes a wideband T/R switch, RF balun and a full phase locked loop (PLL) filter assembly, and it is fabricated in the 0.13 μm SiGe BiCMOS process.
  20. Based on the 0.18um CMOS process, Zheng et al. [49] proposed a burst mode super-regenerative low power UWB generator with OOK modulation, which can be applied in wireless body area network, as shown in Figure 7 (a).
  21. In order to achieve low power consumption, Mercier et al. [50] proposed a low power all-digital UWB pulse generator with BPSK+PPM modulation based on the 90 nm CMOS process, as shown in Figure 7 (b).
  22. As shown in Figure 7 (c), Bourdel et al. [51] proposed a low power pulse response filter UWB pulse generator with OOK-modulated, which is fabricated in the 0.13um CMOS process.
  23. Obviously, the 0.18 μm CMOS process is the most mainstream process at present.
  24. Due to the application of the SiGe BiCMOS process, the power consumption and area in Oliver [47] are 791 pJ/pause and 25 mm2, respectively, which are much larger than other pulse generators.
  25. The pulse signal is transmitted through an environmental channel, which is demodulated and filtered for noise by the receiver.
  26. Maximum ratio combination (MRC) is known as an optimal linear combination technology, which combines all paths with different weights.
  27. The performance of the proposed receiver relationship with the number of equalizer taps and rake fingers is studied by a semi-analytical approach and Monte-Carlo simulations.
  28. As shown in Figure 9 (b), the quadrature analog correlation architecture is used in the proposed receiver which has low energy consumption by reducing the ADC sampling speed.
  29. In a narrowband interference channel, Anis et al. [62] proposed a UWB receiver architecture, which can extract effective signals by narrow-band-pass filters.
  30. The narrow-band-pass filter is consisted of a super regenerative receiver (SRR), as shown in Figure 9 (c).
  31. Based on the 0.13 μm CMOS process, Helleputte et al. [64] proposed an integrated ultra-low power analog front end architecture for UWB pulse radio receivers, as shown in Figure 9 (e).
  32. Local oscillayor is adopted to reduce the power consumption of pulse correlation, and the proposed receiver has a power consumption of 2.7 mW at 39.0625 Mpulses/s pulse rate.
  33. The architecture proposed by Anis [62] has a high bandwidth, which can transmit highspeed signals in indoor situation.
  34. Due to the sub-1 GHz operating bandwidth, the signal transmitted by the architectures [63,64] is the low-speed signal with a strong penetrating ability and wide range.
  35. During the data transmission process, the data are disturbed by noise to cause errors.
  36. In UWB transmitting link, the decoder module is composed of a Viterbi decoder and an RS decoder.
  37. Compared with the traditional architecture, the critical path of ACSU can be reduced by 15% via the proposed architecture.
  38. Compared with traditional MSB ACSU, the proposed decoder saves the area by 12% and increases the speed by 9%.
  39. The look-ahead technique is improved by changing the calculation of ACSU latency to in-crease logarithmically.
  40. In order to simplify BMP, Liu et al. [72] proposed an overall low-complexity BMP architecture based on balanced a binary grouping (BBG) algorithm which can be used to eliminate redundancy and achieve minimum complexity and latency.
  41. The latency and complexity of the ACSU for the lookahead architecture are summarized in Table 4, which represent the number of clock cycles and adders, respectively.
  42. Compared with the conventional method, these proposed architectures are greatly optimized for latency by slightly increasing the complexity.
  43. The architecture proposed by Cheng et al. [71] reduces the complexity by improving the look-ahead method.
  44. In addition, the BBG method is applied to the proposed architecture by Kong et al. [70], which can significantly optimize latency and complexity.
  45. RS decoder is mainly divided into three parts, the calculation of correction factors, the solution of the key equation and the determination of error location and size [74], as shown in Figure 11 (a).
  46. The solution of the key equation is the most complex part of the decoder, which can be implemented by Berlekamp-Massey (BM) algorithm or Euclidean algorithm.
  47. In addition, Sarwate et al. [77] proposed a new reconfigurable invert-free decoding architecture called RiBM, as shown in Figure 11 (b).
  48. The proposed architecture is consisted by one processing element and a compensation unit, which can effectively reduce the hardware complexity.
  49. The process of fractional folding is shown in Figure 11 (e). The number of processing units can be reduced to one, resulting in an ultra-low hardware complexity.
  50. The proposed architecture can reduce the total gate count by 15% more than the ME architecture.
  51. In the proposed architecture, the new initial conditions and polynomials are used, which can significantly reduce the complexity.
  52. Obviously, the mainstream fabrication process of the RS decoders is the 0.18 μm CMOS technology.

Reviewer 3 Report

This paper reported the recent development of UWB physical layer, including encoders, pulse generators, receivers, and decoders. In addition, the detailed comparison and discussion of UWB physical layer are presented to provide suggestions for the design of a high-performance UWB physical layer.

This paper did a comprehensive study on the hot topic. However, professional or native proofreading is strongly suggested. The typos found by the reviewer are listed as follows.

1.       ” UWB physical layer is used to create data information, which is consisted by (‘consist’ is not used in the passive, you can replace them by ‘consists of ’) transmit and receive links.” By the way, there are many “be consisted of (by)” in this article, which are suggested to be replaced by “be made of ”, “be composed of”, etc.

2.       “the transmit link includes (an) encoder and (a) pulse shaping (filter), while the receive link is consisted by (a) receiver and (a) decoder.”

3.       “Finally, the outlook of  UWB physical layer is presented, and its development direction mainly (includes) high speed, low power consumption and less hardware resource”.

4.       Due to the advantages of low power, small size, and strong penetration, (the) UWB system has been widely applied in many fields such as indoor positioning (,) [4-7] wireless body

5.       area network [8-10] and (,) radar(,) [11,12] etc.”

6.       “Specially, UWB physical layer can achieve the data transmitting, which is an important part in UWB system. Therefore, the investigation for (of) UWB physical layer is essential”

7.       “The UWB physical layer transceiver link structure is as show (shown) in Figure 1”

8.       “The encoder is composed by (of) Reed-Solomon (RS) encoder, convolutional encoder, and scrambler.” Please replace the remaining “composed by” by “composed of”

9.       “RS encoder (encoders) are used for payload bit from the physical layer to increase the anti-jamming ability of data.”

10.   “The anti-multipath capability of (the) receiver should be improved”

11.   “The encoder is used to add (a) check bit for data”

12.   “Ren et al. [24] proposed a bit parallel multiplication RS encoder based on (a/the) dual basis.”

13.   “ the number of XOR gates in different algorithms is shown in the (delete) Table 1”

14.   “The convolutional encoder is used to sort RS encoded data, (delete ,) and suppress spectrum” needs to be revised as” The convolutional encoder is used to sort RS encoded data and suppress spectrum”

15.   “pseudo-chaotic jump time (PCTH) technique has been used in convolutional coder (coders) to suppress spectral lines and reduce bit error rate”

16.   “Mo et al. [33] proposed a scrambler with (a) two-layer LFSR to increase the randomness of (the) scrambler”

17.   “Kouassi et al. [34] proposed a UWB transmitter with (a) random pulse-width scrambling method”

18.   (A pulse) Pulse generator is used to load encoded data onto the carrier”

19.   (The) UWB system may conflict with other narrowband systems”

20.   “The on-chip balun with high-pass characteristic (characteristics) has high suppression to PSD in (the) Global Positioning System (GPS) band”

21.   “a new on-off voltage-controlled ring oscillator (VCRO) with (a) complementary switch mode was also proposed”

22.   “Based on (a/the) 0.18um CMOS process”

23.   “Obviously, (the) 0.18 μm CMOS process is the most mainstream process at present”

24.   “Due to the application of (the) SiGe BiCMOS process, the power consumption and area in Oliver [43] (are) 791 pJ/pause and 25 mm2” needs to be revised as”

25.   “The pulse signal is transmitted through (an) environmental channel”

26.   “which combines all paths with different weights.”

27.   “The performance of the proposed receiver relationship with the number of equalizer taps and rake fingers is studied by a semi-analytical approach and Monte-Carlo simulations”

28.   “As showed (shown) in Figure 9 (b)”

29.   “Anis et al. [57] proposed an (a) UWB receiver architecture”

30.   “The narrow-band-pass filter is consisted by (of a) super regenerative receiver (SRR)”

31.   “Local oscillayor is adopted to reduce the power consumption of pulse correlation, and the proposed receiver has the (a) power consumption (of) 2.7 mW at 39.0625 Mpulses/s pulse rate”

32.   “The architecture proposed by Anis [57] has the (a) high bandwidth, which can transmit high-speed signals in indoor situation.”

33.   “the signal transmitted by the architectures [58,59] is the low-speed signal with (a) strong penetrating ability and wide range”

34.   “the data is (are) disturbed by noise to cause errors.”

35.   “the decoder module is composed by (of a) Viterbi decoder and (an) RS decoder”

36.   “Compared with the traditional architecture, the critical path of ACSU can be reduced with (by) 15% by (via) the proposed architecture”

37.   “Compared with traditional MSB ACSU, the proposed decoder saves (the) area of (by) 12% and increases (the) speed of (by) 9%”

38.   “The look-ahead technique is im- 353 proved by changing the calculation of ACSU latency to increases (increase) logarithmically”

39.   “Liu et al. [67] proposed an overall low-complexity BMP architecture based on balanced a binary grouping (BBG) algorithm which can be used to eliminate redundancy and achieve minimum complexity and latency”

40.   “which represent the number of clock cycles and adders, respectively”

41.   “Compared with the conventional method”

42.   “The architecture proposed by Cheng et al. [66] reduces the complexity by improves (improving) the look-ahead method”

43.   “the BBG method is applied to the proposed architecture by Kong et al. [65]”

44.   “the solution of the key equation and the determination of error location and size [69], as shown in Figure 11 (a).”

45.   “The solution of the key equation is the most complex part of the decoder”

46.   “as shown in Figure 11 (c)”

47.   “The proposed architecture is consisted by one processing element and a compensation unit”

48.   “resulting in an ultra-low hardware complexity”

49.   “The proposed architecture can reduce 446 the total gate count by 15% more than the ME architecture”

“In the proposed architecture” 

Author Response

Dear reviewer and editor,

Thanks for the suggestive advice from reviewers and editor. The manuscript has been carefully checked again, and the English language has been improved. The followings are the response to the comments from the reviewers and editor. The corresponding modifications have been made and the important corrections are highlighted in YELLOW in the revised manuscript.

This paper reported the recent development of UWB physical layer, including encoders, pulse generators, receivers, and decoders. In addition, the detailed comparison and discussion of UWB physical layer are presented to provide suggestions for the design of a high-performance UWB physical layer.

Q1. This paper did a comprehensive study on the hot topic. However, professional or native proofreading is strongly suggested. The typos found by the reviewer are listed as follows.

Response: Thanks for the reviewer’s suggestive comments. The manuscript has been carefully checked again, and the English language has been improved. In the revised manuscript, some phrases and sentences have been modified. Thanks!

The modifications are listed as follows:

  1. Finally, the outlook of UWB physical layer is presented, and its development direction mainly includes high speed, low power consumption and less hardware resource.
  2. Due to the advantages of low power, small size, and strong penetration, the UWB system has been widely applied in many fields such as indoor positioning, [4-5,6,7,8 ] wireless body area network [9-10,11,12 ] and, radar, [13-14,15] etc.
  3. Therefore, the investigation of UWB physical layer is essential.
  4. The UWB physical layer transceiver link structure is as shown in Figure 1 [16]. RS encoders are used for payload bit from the physical layer to increase the anti-jamming ability of data [17].
  5. The anti-multipath capability of the receiver should be improved;
  6. The encoder is used to add a check bit for data, which can improve its anti-interference ability in channel transmission.
  7. In order to deal with this problem, Ren et al. [27] proposed a bit parallel multiplication RS encoder based on the dual basis, as shown in Figure 2 (a).
  8. In the last years, pseudo-chaotic jump time (PCTH) technique has been used in convolutional coders to suppress spectral lines and reduce bit error rate (BER).
  9. Mo et al. [36] proposed a scrambler with a two-layer LFSR to increase the randomness of the scrambler.
  10. In addition, Kouassi et al. [37] proposed a UWB transmitter with a random pulse-width scrambling method.
  11. A pulse generator is used to load encoded data onto the carrier, which is essential in UWB physical layer.
  12. The UWB system may conflict with other narrowband systems, so many circuits have been proposed to reduce PSD [38,39].
  13. The on-chip balun with high-pass characteristics has high suppression to PSD in the Global Positioning System (GPS) band.
  14. Also, Sim et al. [41] proposed a low power, high peak value UWB pulse generator based on the 0.18 μm process for 6-10 GHz, which is consisted of a pulse generator, a pulse shaping filter and a pulse oscillator, as shown in Figure 5. (b).
  15. In addition, a new on-off voltage-controlled ring oscillator (VCRO) with a complementary switch mode was also proposed, which can reduce power consumption by avoiding generating base-band energy.
  16. Based on the 90 nm CMOS process, Hedayati et al. [43] proposed a fully integrated analog UWB pulse transmitter with BPSK modulation, as shown in Figure 5. (d), which can coexist with IEEE 802.11a system.
  17. In order to solve this problem, Choi et al. [45] proposed an all-digital pulse generator based on the 0.13 μm CMOS technology, consisting of a delay line structure, as shown in Figure 6 (a).
  18. Based on the 65 nm CMOS process, Na et al. [46] proposed an all-digital UWB pulse generator with three optional channels.
  19. As shown in Figure 6 (b), the proposed pulse generator includes a wideband T/R switch, RF balun and a full phase locked loop (PLL) filter assembly, and it is fabricated in the 0.13 μm SiGe BiCMOS process.
  20. Based on the 0.18um CMOS process, Zheng et al. [49] proposed a burst mode super-regenerative low power UWB generator with OOK modulation, which can be applied in wireless body area network, as shown in Figure 7 (a).
  21. In order to achieve low power consumption, Mercier et al. [50] proposed a low power all-digital UWB pulse generator with BPSK+PPM modulation based on the 90 nm CMOS process, as shown in Figure 7 (b).
  22. As shown in Figure 7 (c), Bourdel et al. [51] proposed a low power pulse response filter UWB pulse generator with OOK-modulated, which is fabricated in the 0.13um CMOS process.
  23. Obviously, the 0.18 μm CMOS process is the most mainstream process at present.
  24. Due to the application of the SiGe BiCMOS process, the power consumption and area in Oliver [47] are 791 pJ/pause and 25 mm2, respectively, which are much larger than other pulse generators.
  25. The pulse signal is transmitted through an environmental channel, which is demodulated and filtered for noise by the receiver.
  26. Maximum ratio combination (MRC) is known as an optimal linear combination technology, which combines all paths with different weights.
  27. The performance of the proposed receiver relationship with the number of equalizer taps and rake fingers is studied by a semi-analytical approach and Monte-Carlo simulations.
  28. As shown in Figure 9 (b), the quadrature analog correlation architecture is used in the proposed receiver which has low energy consumption by reducing the ADC sampling speed.
  29. In a narrowband interference channel, Anis et al. [62] proposed a UWB receiver architecture, which can extract effective signals by narrow-band-pass filters.
  30. The narrow-band-pass filter is consisted of a super regenerative receiver (SRR), as shown in Figure 9 (c).
  31. Based on the 0.13 μm CMOS process, Helleputte et al. [64] proposed an integrated ultra-low power analog front end architecture for UWB pulse radio receivers, as shown in Figure 9 (e).
  32. Local oscillayor is adopted to reduce the power consumption of pulse correlation, and the proposed receiver has a power consumption of 2.7 mW at 39.0625 Mpulses/s pulse rate.
  33. The architecture proposed by Anis [62] has a high bandwidth, which can transmit highspeed signals in indoor situation.
  34. Due to the sub-1 GHz operating bandwidth, the signal transmitted by the architectures [63,64] is the low-speed signal with a strong penetrating ability and wide range.
  35. During the data transmission process, the data are disturbed by noise to cause errors.
  36. In UWB transmitting link, the decoder module is composed of a Viterbi decoder and an RS decoder.
  37. Compared with the traditional architecture, the critical path of ACSU can be reduced by 15% via the proposed architecture.
  38. Compared with traditional MSB ACSU, the proposed decoder saves the area by 12% and increases the speed by 9%.
  39. The look-ahead technique is improved by changing the calculation of ACSU latency to in-crease logarithmically.
  40. In order to simplify BMP, Liu et al. [72] proposed an overall low-complexity BMP architecture based on balanced a binary grouping (BBG) algorithm which can be used to eliminate redundancy and achieve minimum complexity and latency.
  41. The latency and complexity of the ACSU for the lookahead architecture are summarized in Table 4, which represent the number of clock cycles and adders, respectively.
  42. Compared with the conventional method, these proposed architectures are greatly optimized for latency by slightly increasing the complexity.
  43. The architecture proposed by Cheng et al. [71] reduces the complexity by improving the look-ahead method.
  44. In addition, the BBG method is applied to the proposed architecture by Kong et al. [70], which can significantly optimize latency and complexity.
  45. RS decoder is mainly divided into three parts, the calculation of correction factors, the solution of the key equation and the determination of error location and size [74], as shown in Figure 11 (a).
  46. The solution of the key equation is the most complex part of the decoder, which can be implemented by Berlekamp-Massey (BM) algorithm or Euclidean algorithm.
  47. In addition, Sarwate et al. [77] proposed a new reconfigurable invert-free decoding architecture called RiBM, as shown in Figure 11 (b).
  48. The proposed architecture is consisted by one processing element and a compensation unit, which can effectively reduce the hardware complexity.
  49. The process of fractional folding is shown in Figure 11 (e). The number of processing units can be reduced to one, resulting in an ultra-low hardware complexity.
  50. The proposed architecture can reduce the total gate count by 15% more than the ME architecture.
  51. In the proposed architecture, the new initial conditions and polynomials are used, which can significantly reduce the complexity.
  52. Obviously, the mainstream fabrication process of the RS decoders is the 0.18 μm CMOS technology.

Round 2

Reviewer 1 Report

The authors responded to all comments. The improved work significantly enriched the described content of the review article.

Reviewer 2 Report

While I do appreciate the authors efforts to accommodate the critique made wit regards to the first version, then all my main reservations still apply to the updated version. The papers intends to be a paper projecting the tends and directions of the field, which is the UWB physical layer. However, it does not achieve this to any satisfactory degree. Instead it comes across as a survey paper that more or less incoherently puts a finger on a few publications. As such the paper brings only limited value, if any, to the scientific community.

As a final comment, and this I have tried to substantiated by my handwritten comments, the paper is still poorly presented and a significant effort is needed just to solve the presentation part .. content aside. 

Reviewer 3 Report

Accepted as it is.